# Optimizing Management Practices under Straw Regimes for Global Sustainable Agricultural Production

**Pengfei Li** [1,2], **Afeng Zhang** [1,2,*], **Shiwei Huang** [1,2], **Jiale Han** [1,2], **Xiangle Jin** [1,2], **Xiaogang Shen** [1,2], **Qaiser Hussain** [3], **Xudong Wang** [1,2], **Jianbin Zhou** [1,2] **and Zhujun Chen** [1,2]

[1]  College of Natural Resources and Environment, Northwest A&F University, Yangling 712100, China
[2]  Key Laboratory of Plant Nutrition and the Agro-Environment in Northwest China, Ministry of Agriculture, Yangling 712100, China
[3]  Institute of Soil and Environmental Sciences, Pir Mehr Ali Shah Arid Agriculture University, Rawalpindi P.O. Box 46300, Pakistan
[*]  Correspondence: zhangafeng@nwsuaf.edu.cn

**Abstract:** Straw input is a helpful approach that potentially improves soil fertility and crop yield to ensure food security and protect the ecological environment. Nevertheless, unreasonable straw input results in massive greenhouse gas (GHG) emissions, leading to climate change and global warming. To explore the optimum combination of straw input and management practices for achieving green agricultural production, a worldwide data set was created using 3452 comparisons from 323 publications using the meta-analysis method. Overall, straw input increased soil carbon and nitrogen components as compared with no straw input. Additionally, straw input significantly boosted crop yield and nitrogen use efficiency (NUE) by 8.86% and 22.72%, respectively, with low nitrogen fertilizer rate benefiting the most. The cumulative of carbon dioxide ($CO_2$), methane ($CH_4$), and nitrous oxide ($N_2O$) emissions increased by 24.81%, 79.30%, and 28.31%, respectively, when straw was added. Global warming potential (GWP) and greenhouse emission intensity (GHGI) increased with the application of straw, whereas net global warming potential (NGWP) decreased owing to soil carbon sequestration. Low straw input rate, straw mulching, application of straw with C/N ratio > 30, long-term straw input, and no-tillage combined with straw input all result in lower GHG emissions. The GWP and GHGI were strongly related to area-scaled $CH_4$ emissions, but the relationship with $N_2O$ emissions was weak. Straw application during the non-rice season is the most important measure for reducing $CH_4$ emissions in paddy–upland fields. An optimum straw management strategy coupled with local conditions can help in climate change mitigation while also promoting sustainable agricultural production.

**Keywords:** straw input; different farming regimes; greenhouse gas emissions; carbon and nitrogen components; meta-analysis; net global warming potential





## 1. Introduction

Agricultural production is a major source of greenhouse gas (GHG) emissions, accounting for approximately 5.0–5.8 Pg carbon dioxide ($CO_2$)-equivalent year$^{-1}$ of anthropogenic GHG emissions [1]. It is estimated that agricultural methane ($CH_4$) and nitrous oxide ($N_2O$) emissions account for at least 50% and 60% of global anthropogenic $CH_4$ and $N_2O$ emissions, respectively [2]. The 4 per mille or 4 per 1000 goal is to increase global soil organic carbon stocks by 0.4% per year in order to mitigate climate change and achieve food security through anthropogenic sources [3]. In this regard, increasing soil organic carbon sequestration is one of the most important strategies for lowering global warming potential (GWP), with a significant potential to mitigate climate change [4,5]. Optimizing management practices contributes to the long-term development of agriculture. Straw input to croplands is a popular and cost-effective management practice in China and around the world due to its ability to improve soil fertility [6], increase crop productivity [7],

and increase soil C sequestration [8]. Annual crop residue production is estimated to be $3.8 \text{ Gt yr}^{-1}$ globally [9]. However, it is noted that straw input may accelerate soil GHG emissions and increase GWP, which may partially offset the benefits of climate change mitigation by increasing soil organic carbon (SOC) sequestration [10,11]. The trade-off of the positive and negative effects of straw input lacks a comprehensive assessment.

Meta-analysis is a comprehensive statistical analysis to assess the consistency of independent experiments involving same subject conclusion [12]. Meta-analysis uses more scientific processes and methods to quantitatively analyze and process a larger number of literature research results, which makes the conclusions closer to the objective facts; and that can also flexibly divide data into different subgroups for analysis according to the factors that may lead to differences, so as to find the causes of differences. Meta-analysis is widely used in numerous scientific fields. In agriculture, this method is widely applied to assess the effects of different agronomic practices on a regional or global scale [13]. Liu et al. conducted a global meta-analysis on the effects of straw input on soil C dynamics and concluded that straw input increased C sink in upland soils but increased C emissions in rice paddy [14]. According to a meta-analysis in China's croplands, straw input significantly increased $CH_4$ emissions by 130.9%, which was the highest, followed by $CO_2$ and $N_2O$ emissions [15]. Previous report proved in a global meta-analysis that nitrogen fertilizer input increased soil $N_2O$ for all nitrogen input forms, and that the effect size of nitrogen input on soil $N_2O$ emissions increased from 62% to 127% with increasing nitrogen input rate, and from 80% to 192% with increasing experimental duration [16]. A meta-analysis conducted straw input significantly increased the crop yield by 8.1% in China [11]. In a global meta-analysis, Xia et al. reported straw input with the same amount of nitrogen fertilizer significantly increased soil organic carbon content (14.9%) compared to nitrogen fertilization [17]. Different straw management measures, such as straw input manner, application rate, straw quality [17–19], climatic conditions (temperature and precipitation), nitrogen fertilizer rate, tillage method, and land use type [20,21] also had significant effects on GHG emissions, soil carbon storage, soil properties, and crop production; these drivers will vary with agricultural-related practices. Therefore, evaluating how GHG emissions, crop production, and soil carbon and nitrogen components respond to different farmland management practices under straw regimes optimizes existing methods for implementing straw input and maximizing profits.

Previous meta-analysis studies generally focused on a single aspect of straw input, such as crop yield [22,23], SOC content [24], and GHG emissions [19,25], but did not address overall environmental impacts [e.g., GWP, greenhouse gas emission intensity (GHGI), and net global warming potential (NGWP)], as well as nitrogen use efficiency (NUE), which hampered. Information derived from a multifaceted and global scale meta-analysis could aid in understanding how soil–crop systems respond to straw input, more practically, in optimizing the straw input-based field management system. Here, we show the results of a global meta-analysis that was based on 323 peer-reviewed publications. The study's objectives were to (a) assess the overall effects of straw input on soil carbon and nitrogen contents, crop production, environmental conditions; (b) investigate the variation characteristics of GHG emissions under the quantity and quality of straw amendment; (c) identify the combined effects of various farming practices and climate conditions with straw input and determine their interactive relationships, optimizing management practices under straw regimes to provide a scientific basis for straw input to improve soil fertility globally; and (d) explore the interactions between GHG emissions and soil carbon storage by analyzing greenhouse gas warming potential and revealing the impact mechanism of straw input on GHG emissions, thereby determining trade-offs between mitigating climate change and maintaining food production.

## 2. Materials and Methods

### 2.1. Data Sources and Compilation

The relevant literature was gathered using databases such as Web of Science, Elsevier, Google Scholar, and the China National Knowledge Infrastructure, with the keywords 'straw return' or 'straw mulching' or 'crop residue' and 'greenhouse gas' or 'soil properties' or 'crop yield' published prior to 2021. The data were compiled from publications that detailed GHG emissions and soil properties of concern and compared the effects of various agricultural management practices on these properties with and without straw amendment. If the results were presented graphically, the data were extracted numerically using the Get Data software. Data were extracted from 323 scientific publications containing a total of 3452 individual observations comparing control (no-straw input treatments) to straw input treatments (Figure 1). The fundamental properties of soils and straw were gathered, as well as descriptions of the crop type and straw variables. All experiments were conducted in the field. Eligible studies are listed in Supplementary Materials Date S1.

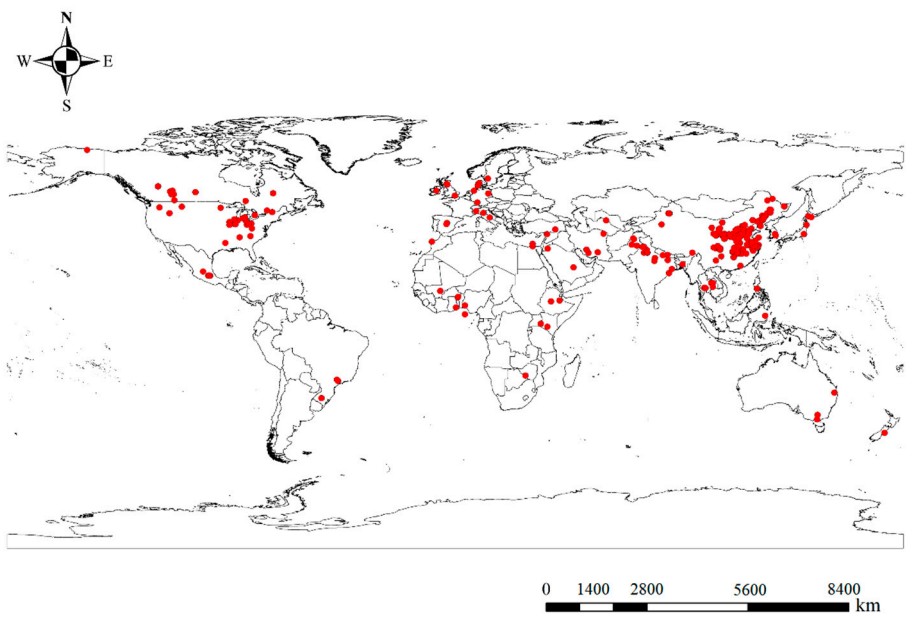

**Figure 1.** Global distribution of study sites included in the database for meta-analysis.

### 2.2. Data Normalization

In each published study, the standard deviation (SD) used as a measure of variance was calculated from the measured variance. When standard errors (SE) were provided, they were converted to standard deviations using the Equation (1):

$$SD = SE \times \sqrt{n} \tag{1}$$

where n is the number of replications.

### 2.3. Meta-Analysis

The magnitude of change in a property (also known as 'effect size') in response to an experimental treatment is estimated using meta-analysis across a wide range of variables [26,27]. The effect of straw application on each index was investigated using weighted integration; in order to improve its accuracy and effectively reduce heterogeneity, it is necessary to consider variation within and between studies, so the effect value (lnR) was chosen as the effect size to reduce bias in meta-analysis and calculated using Equation (2):

$$\ln R = \ln (X_t / X_c) = \ln (X_t) - \ln(X_c) \tag{2}$$

where Xt and Xc are means of the variable in straw treatment and control groups, respectively. Its variance (v) was calculated using Equation (3):

$$U = \frac{S_t^2}{n_t x_t^2} + \frac{S_c^2}{n_c x_c^2} \tag{3}$$

where $n_t$ and $n_c$ are the sample sizes for the treatment and control groups, respectively, and $S_t$ and $S_c$ are the standard deviations for the treatment and control groups, respectively. Meta Win 2.0 was used to compute mean effect sizes and 95% bootstrap confidence intervals (CIs). The CIs calculated by bootstrap method is wider than the standard confidence interval [28]. In categorical meta-analysis, the CIs of lnR were assessed using bootstrapping with 4999 iterations, and randomization procedures with 4999 replications were used to test the between-group heterogeneity [29]. The response ratio and CIs of the treatments presented were converted to percent change from lnR [30]. The straw treatment was considered significant if the CIs of the response ratio in each figure did not overlap with zero. Responses from different groups were considered distinct if their CIs did not overlap. To make it easier for readers to understand, we back-transformed the results as a percentage change (Y) using Equation (4):

$$Y = (e^{lnR} - 1) \times 100\%. \tag{4}$$

In subgroup analysis, the difference between groups is considered significant if the bootstrap confidence intervals of Y values in different groups do not overlap; otherwise, the difference between groups is not significant. For analyzing the combined effects of straw input under different farming practices and understanding the changes of Y, possible variables, such as climatic conditions, straw quantity and quality, nitrogen fertilizer rate, straw input duration, straw application method, tillage method, and land use type were selected. Categorical meta-analysis was conducted to assess the difference between subgroups of climatic conditions (mean annual temperature and mean annual precipitation); straw quantity (low straw incorporation or mulching rate (IR, MR < 4.5 t ha$^{-1}$), middle incorporation or mulching rate (4.5 ≤ IR, MR < 9 t ha$^{-1}$), high incorporation or mulching rate (IR, MR ≥ 9 t ha$^{-1}$); straw quality (straw C/N ratio <30, 30–60, ≥60); straw input duration (short-term input (<5 year), long-term input (≥5 year); nitrogen fertilizer rate (<120, 120–240, and 240 kg N ha$^{-1}$); tillage method (traditional tillage, no-tillage); and land use type (upland field, paddy field, and paddy–upland rotation).

Apparent nitrogen use efficiency (NUE) (kg N ha$^{-1}$) was calculated using Equation (5):

$$NUE = (NU_T - NU_{CK})/F \tag{5}$$

where $NU_T$ and $NU_{CK}$ were the crop nitrogen uptake levels for the fertilizer treatment and the CK treatment without nitrogen fertilizer input as control, respectively (kg N ha$^{-1}$), and F was the amount of nitrogen fertilizer applied.

Due to the differences in the warming effects of $N_2O$ and $CH_4$ emissions, GWP (kg $CO_2$-eq ha$^{-1}$) and GHGI (kg $CO_2$-eq kg$^{-1}$) were used as indicators to evaluate the comprehensive effects of RR on total GHGs emissions [31]. The GWP of $N_2O$ and $CH_4$ was estimated using Equation (6):

$$GWP_{100yr} \text{ (kg } CO_2\text{-eq ha}^{-1}) = RN_2O \times 298 + RCH_4 \times 25 \tag{6}$$

where $GWP_{100yr}$ is the comprehensive warming potential on a 100-year time scale, $RN_2O$ and $RCH_4$ are seasonal cumulative emissions of $N_2O$ and $CH_4$ (kg ha$^{-1}$), respectively. The GHGI was calculated with the following Equation (7):

$$GHGI \text{ (kg } CO_2\text{-eq kg}^{-1} \text{ grain yield)} = GWP/\text{grain yield}. \tag{7}$$

The net global warming potential (NGWP, kg $CO_2$-eq ha$^{-1}$) was estimated using Equations (8) and (9):

$$NGWP \text{ (kg } CO_2\text{-eq ha}^{-1}) = GWP - SOCSR \times 44/12 \tag{8}$$

$$SOCSR = (SOCD_t - SOCD_0)/T \tag{9}$$

$SOCD_t$ and $SOCD_0$ refer to SOC storage in the final and first years, respectively, and T denotes experiment duration.

The regression technique was used for fitting, and finally the linear or non-linear relationship between the independent variable and the dependent variable was determined according to the goodness-of-fit test translation. All statistical analyses were performed using SPSS 26.0.

## 3. Results

### 3.1. Impacts of Straw Input on Soil Carbon and Nitrogen Components

Straw input had significantly positive effects on soil carbon and nitrogen components as compared to control. Straw input increased significantly SOC (13.25%), soil dissolved organic carbon (DOC, 27.93%), soil microbial biomass carbon (MBC, 27.84%), and increased soil total nitrogen (TN, 13.1%), soil available nitrogen (SAN, 5.96%), soil microbial biomass nitrogen (MBN, 32.98%), soil ammonium nitrogen content ($NH_4^+$-N, 6.27%), and soil nitrate nitrogen ($NO_3^-$-N, 6.94%; Figure 2a–h). The increment of SOC content under straw input was significantly related to mean annual temperature (MAT; $p < 0.001$; Table 1), which increased the most when MAT ≥ 20 °C. Straw input rate ($p < 0.001$) and nitrogen fertilizer rate ($p < 0.05$), which enhanced with the rate of straw incorporation, straw mulching, and nitrogen fertilizer application increasing (Figure 2a). The increment of SOC content increased the most when MAT ≥ 20 °C. For straw quality, straw C/N ratio ($p < 0.01$) and input duration ($p < 0.001$) were both positively correlated with the increment of SOC content. The increment of SOC is greater under straw C/N ratio ≥ 60 (14.49%) than 30–60 (11.18%) and <30 (8.74%), respectively. Additionally, the increment of SOC was significantly influenced by land use type ($p < 0.01$; Table 1), and the SOC content increased by 14.49%, 13.53% and 10.08% under upland fields, paddy fields, and paddy–upland rotation under straw input, respectively (Figure 2a). In our synthesis, soil labile C components (eg, DOC and MBC) showed significantly positive responses under straw input, and varied with other factors. The increment of DOC was correlated with nitrogen fertilizer rate ($p < 0.01$), MAT, straw input rate, straw C/N ratio, and land use type (both $p < 0.05$); however, the increment of MBC was correlated with nitrogen fertilizer rate ($p < 0.01$) and MAP ($p < 0.05$; Table 1). Specifically, the highest MBC content was observed at MAP with ≥1200 mm (46.33%), then MAP with 600–1200 mm (31.90%); however, there was no significant difference at MAP with <600 mm under straw input, respectively (Figure 2c). Straw input rate had a similar trend for DOC and MBC content change, and middle and high straw input rate (≥4.5 t ha$^{-1}$) significantly increased DOC and MBC, but not for low straw input rate (<4.5 t ha$^{-1}$) treatment.

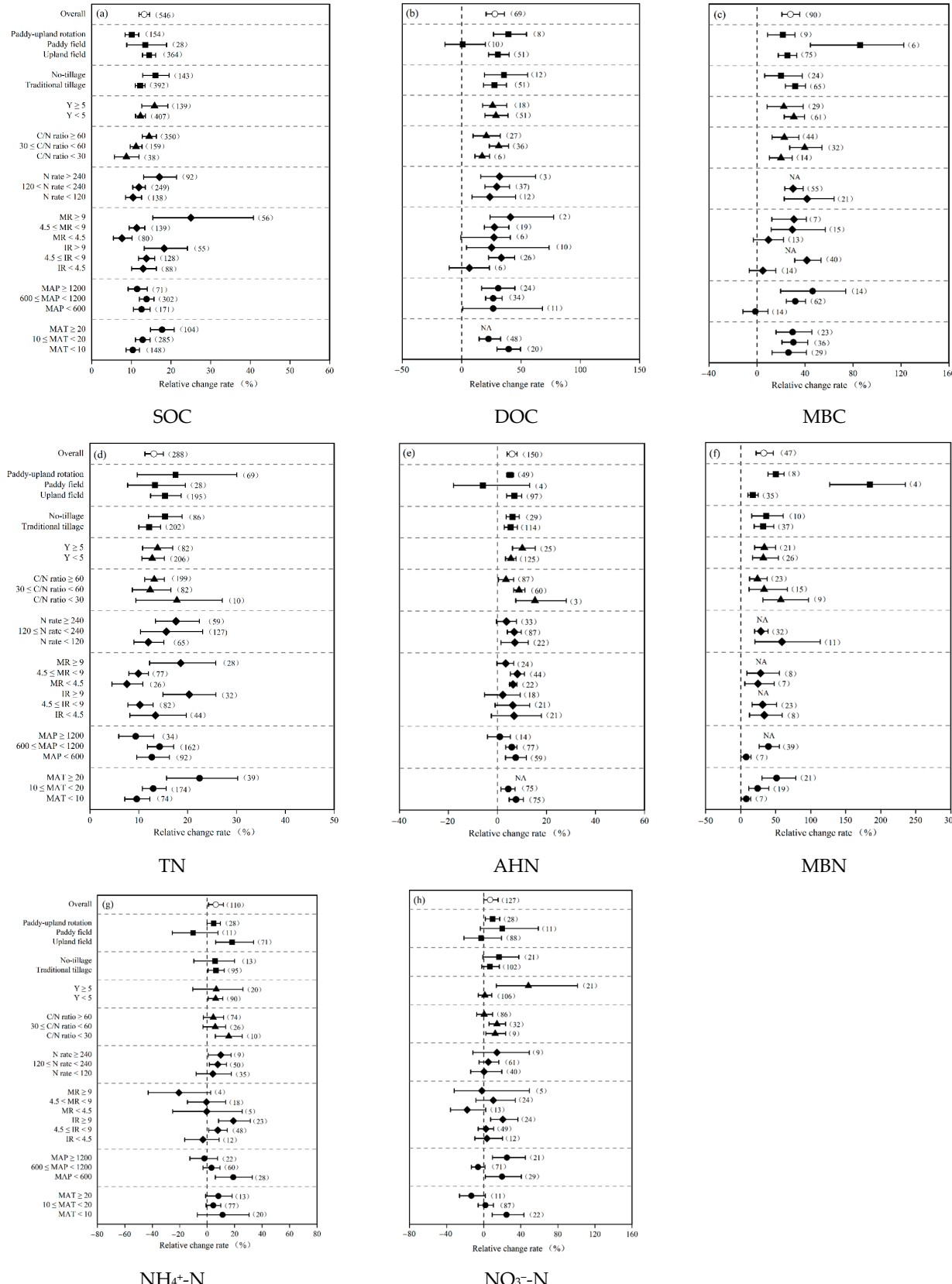

**Figure 2.** Changes in soil carbon and nitrogen components under straw input to cropland. SOC, soil organic carbon; DOC, dissolved organic carbon; MBC, microbial biomass carbon; soil TN, soil total nitrogen; AHN, alkali-hydrolyzale nitrogen; and MBN, microbial biomass nitrogen. MAT denotes

mean annual temperature; MAP denotes mean annual precipitation; IR represents straw incorporation rate, MR represents straw mulching rate (t ha$^{-1}$); N rate denotes the application rate of nitrogen fertilizer (kg N ha$^{-1}$); and Y < 5 denotes that straw was continuously used for less than 5 years. Parentheses numbers indicate the number of observations, and error bars represent 95% bootstrap confidence intervals. The effect was considered significant if the 95% CIs of the mean effect did not overlap with zero.

**Table 1.** Linear regression analysis between soil properties, crop productivity, and greenhouse gases emission with different climate conditions and farmland management practices under straw input to global cropland.

| Parameters | Effect size of variables | | | | | | | | | | | | |
|---|---|---|---|---|---|---|---|---|---|---|---|---|---|
| | SOC | DOC | MBC | TN | AHN | MBN | NH$_4^+$-N | NO$_3^-$-N | Yield | NUE | CO$_2$ Emission | CH$_4$ Emission | N$_2$O Emission |
| MAT | *** | *- | ns | *** | *- | * | ns | ns | * | ns | ns | ns | ns |
| MAP | ns | ns | * | ns | ns | * | ns | ns | ns | ns | * | * | ns |
| Straw input manner | ns | ns | ns | ns | ns | ns | * | ns | ns | * | ns | ns | ns |
| Straw input rate | *** | * | ns | *** | ns | * | * | * | ** | *- | * | **- | ** |
| Nitrogen fertilizer rate | * | ** | **- | ** | ns | ns | ns | ns | **- | *- | ns | ns | ns |
| Straw C/N ratio | ** | *- | ns | *- | *- | ns | ns | ns | ns | *- | ns | ns | *- |
| Straw input duration | *** | ns | ns | ns | ns | ns | ns | * | ns | ns | *- | *- | ns |
| Tillage method | * | ns | ns | ns | ns | ns | ns | ns | ns | ns | * | *** | ns |
| Land use type | ** | ns | ns | ** | ns | * | ns | ns | ns | ns | * | * | * |

Abbreviations: MAT, mean annual temperature; MAP, mean annual precipitation; SOC, soil organic carbon; DOC, dissolved organic carbon; MBC, microbial biomass carbon; TN, soil total nitrogen; AHN, alkali-hydrolyzable nitrogen; MBN, microbial biomass nitrogen; and NUE, nitrogen use efficiency. The symbols *, **, and *** denote corrected significance at $p < 0.05$, $p < 0.01$, and $p < 0.001$, respectively, ns denotes the linear relationship is not significant, and - denotes a negative linear relationship.

Soil TN and nitrogen components were significantly increased under straw input combined with other regime practices, and also influenced by climate conditions. Straw input effect on TN content was dependent on the MAT, straw input rate, nitrogen fertilizer rate, land use type, and straw C/N ratio (Table 1). Additionally, the MAP and straw C/N ratios were negatively correlated with the increment of AHN content ($p < 0.05$). In addition, the increment of MBN, NH$_4^+$-N, and NO$_3^-$-N content significantly increased with straw input rate (both $p < 0.05$; Table 1). The increment of MBN content also significantly increased with the increase in MAT and MAP, and land use type significantly regulated the effects of straw input on MBN content, which significantly increased by 17.23%, 184.74%, and 50.05% for upland field, paddy field, and paddy–upland rotation, respectively (Figure 2f). In addition, the increment of NH$_4^+$-N content was correlated with straw input manner ($p < 0.05$), and soil NH$_4^+$-N content was increased under straw incorporation treatment but not for straw mulching. Straw input duration was correlated with the increment of NO$_3^-$-N content ($p < 0.05$; Table 1), which increased by 48.25% under long-term straw input, but not for short-term straw input (Figure 2g–h).

### 3.2. Effects of Straw Input on Crop Yield and Nitrogen Use Efficiency

Straw amendment increased crop yield by 8.86% as compared to no-straw amendment. Crop yield increases were significantly positively correlated with MAT ($p < 0.05$) and straw input rate ($p < 0.01$), but significantly negatively correlated with nitrogen fertilizer rate ($p < 0.01$; Table 1), which was observed to enhance with increasing straw incorporation rate; however, the maximum increase in crop yield was observed with straw mulching rate ranging from 4.5 to 9 t ha$^{-1}$, but not at $\geq 9$ t ha$^{-1}$. Additionally, straw C/N ratio, straw input duration, tillage method, and land use type had no significant effect on crop yield (Figure 3a). Straw input increased NUE by 22.72%. The variation in NUE was significantly and negatively related to straw input rate, nitrogen fertilizer rate, and straw C/N ratio (both $p < 0.05$; Table 1). The greatest increase in NUE was observed under straw incorporation combined with low nitrogen fertilizer rate treatment (<120 kg N ha$^{-1}$), but

not for middle nitrogen fertilizer rate (120–240 kg N ha$^{-1}$) and high nitrogen fertilizer rate ($\geq$240 kg N ha$^{-1}$) treatments (Figure 3b).

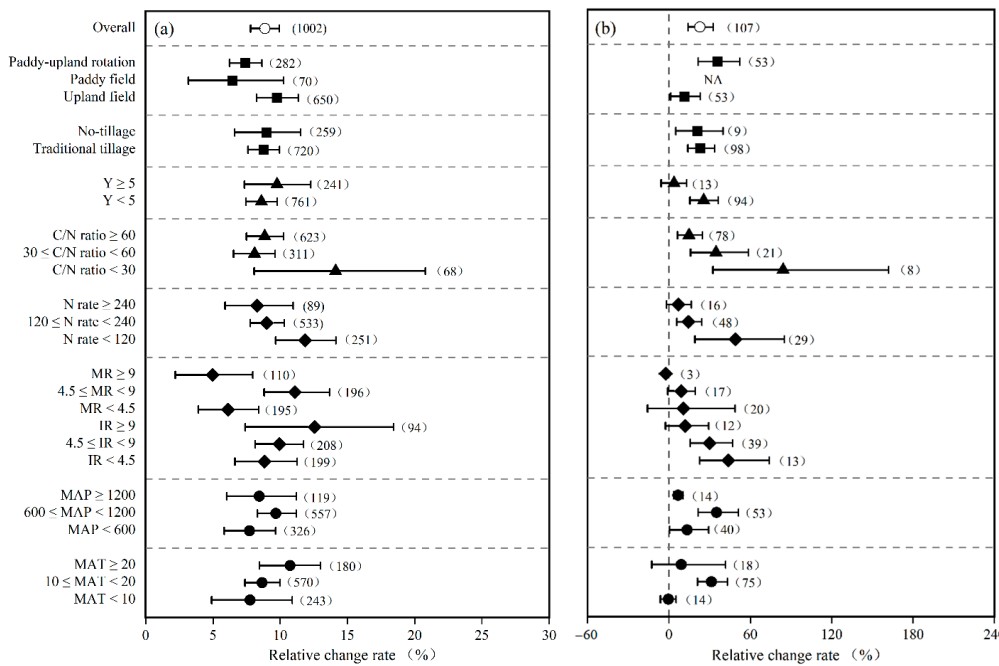

**Figure 3.** Changes in (**a**) crop yield and (**b**) nitrogen use efficiency under straw input to cropland.

### 3.3. Impacts of Straw Input on Greenhouse Gases Emissions

Overall, straw input increased cumulative $CO_2$ emissions by 24.81% as compared to no straw input (Figure 4a), and the effect was dependent on the MAP, straw input rate, tillage method and land use type, and straw input duration (all $p < 0.05$; Table 1). Furthermore, the cumulative $CO_2$ emissions showed strong regression relationships with straw input rate, which was increased as the straw input rate increased for all straw types (Figure 5). The largest increase in cumulative $CO_2$ emissions occurred when MAT was $\geq$20 °C, or MAP was $\geq$1200 mm. Furthermore, under straw input, traditional tillage resulted in significantly higher $CO_2$ emissions than no-tillage. The increase in cumulative $CO_2$ emissions was observed increased by 49.98%, 24.25%, and 4.38% under paddy–upland rotation, upland field, and paddy field, respectively (Figure 4a). In addition, the cumulative $CO_2$ emissions were affected by straw input duration, which increased by 29.74% and 14.26% under the short-term and long-term addition, respectively (Figure 4a).

The cumulative $CH_4$ emissions increased by 79.30% under straw input as compared to no straw input (Figure 4b), which was significantly related to MAP, land use type (both $p < 0.05$), and tillage method ($p < 0.001$), but significantly decreased with straw input rate ($p < 0.01$) and input duration ($p < 0.05$; Table 1). The cumulative $CH_4$ emissions showed significant regression relationship with rice and Triticeae straw input rate, it can be observed that cumulative $CH_4$ emissions generally decreased and then increased with increasing rice straw input rate, while cumulative $CH_4$ emission decreased with increasing Triticease straw input rate. In addition, there is a significant regression relationship between nitrogen fertilizer rate and $CH_4$ emissions under rice and Triticeae straw input (Figure 5). The increase in cumulative $CH_4$ emissions was 211.90% in the paddy field and 92.90% in the paddy–upland rotation field, but there was no significant increase in the upland field. No-tillage and long-term input duration had no significant effect on $CH_4$ emissions under straw input, whereas traditional tillage and short-term input treatment significantly increased $CH_4$ emissions by 117.43% and 112.74%, respectively (Figure 4b). The cumulative $CH_4$ emissions were also affected by straw C/N ratio, which increased by 122.58%, 45.48%, and 41.18% under straw C/N ratio at 30–60, <30, and $\geq$60, respectively (Figure 4b).

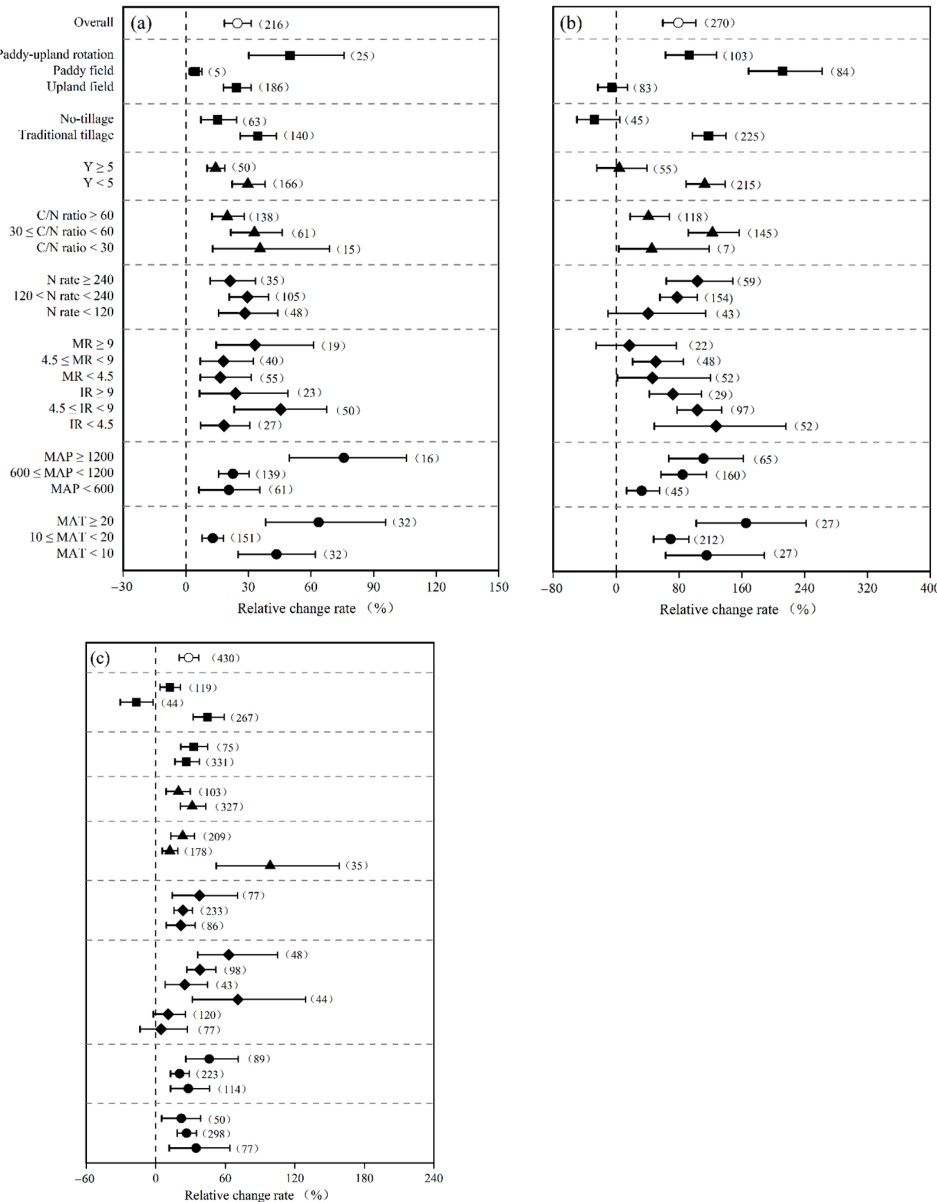

**Figure 4.** Changes in cumulative (**a**) $CO_2$, (**b**) $CH_4$, and (**c**) $N_2O$ emissions under straw input to cropland.

Straw input increased cumulative $N_2O$ emissions by 28.31% when compared to no-straw input (Figure 4c). There was a significant relationship between straw input rate ($p < 0.01$), straw C/N ratio ($p < 0.05$), land use type ($p < 0.05$), and $N_2O$ emissions (Table 1). The effect of straw input rate and nitrogen fertilizer rate on $N_2O$ emissions depended on straw types. Our results indicated a significantly positive regression relationship between cumulative $N_2O$ emissions and Triticeae, Fabaceae straw input rate (Figure 5). In addition, a significant regression relationship between cumulative $N_2O$ emissions and nitrogen fertilizer rate under maize, rice, and Fabaceae straw addition was also shown in Figure 5. The cumulative $N_2O$ emissions increased with increased nitrogen fertilizer rate under maize and rice straw input, but it decreased and then increased with the increase in nitrogen fertilizer rate under Fabaceae straw input. The cumulative $N_2O$ emissions varied with straw C/N ratio, which was increased by 98.81%, 12.12%, and 23.20% under straw C/N ratio at <30, 30–60, and ≥60, respectively. In terms of land use type, straw input significantly increased $N_2O$ emissions in both upland and paddy–upland rotation, but decreased $N_2O$ emissions in paddy fields (Figure 4c).

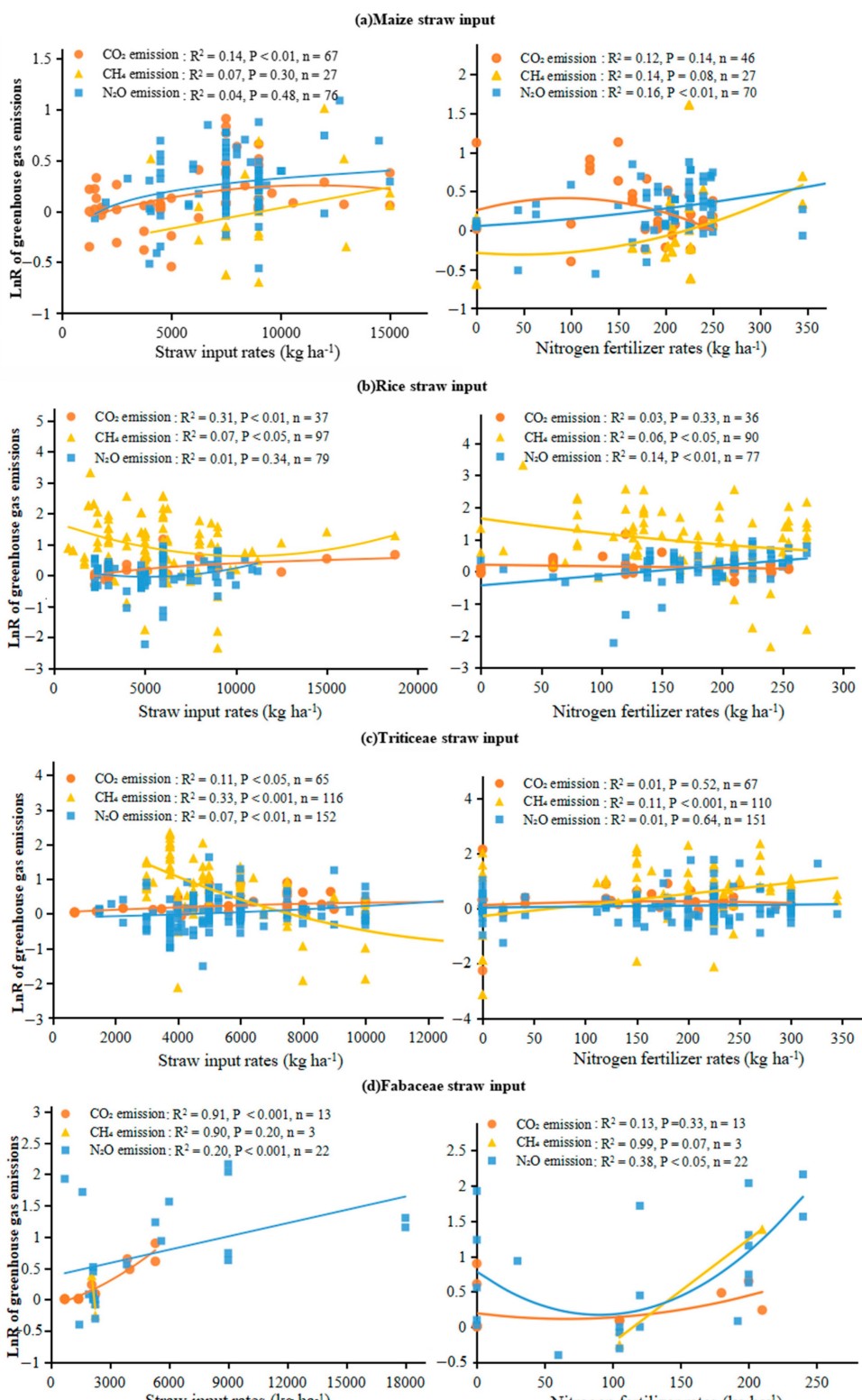

**Figure 5.** Relationship between straw input rates, nitrogen fertilizer rates, and greenhouse gas emissions for different straw types.

### 3.4. Impacts of Straw Input on Global Warming Potential, Greenhouse Emission Intensity and Net Global Warming Potential

Overall, straw input increased GWP by 55.18% as compared to no-straw input (Figure 6a). Particularly, the response of GWP to straw incorporation (81.55%) was significantly higher

than straw mulching (18.93%). The paddy field (91.02%) and paddy–upland rotation field (65.17%) had a greater positive effect than that of the upland field (23.69%), owing to increased CH$_4$ emissions from paddy soil under straw application. Furthermore, traditional tillage increased GWP by 70.28%, but not no-tillage treatment (Figure 6a). Straw input significantly increased GHGI by 46.40% as compared to no-straw input (Figure 6b). The GHGI increased by 66.13%, 120.56%, 48.56%, 52.91%, and 59.07% under straw incorporation, paddy field, paddy–upland rotation system, short-term straw input, and traditional tillage, respectively. However, the NGWP was reduced by 28.29% under straw amendment as compared to no-straw input (Figure 6c), which was reduced by 25.90% and 32.04% under straw incorporation and mulching, respectively. And the NGWP was decreased by 29.85%, 47.08%, 31.09%, and 27.62% under MAT was 10–20 °C, MAP < 600 mm, middle, and high nitrogen fertilizer rate, respectively, but no significant effect at low nitrogen fertilizer rate. In terms of land use type, straw input significantly reduced NGWP by 43.40% in upland field and 24.49% in a paddy–upland rotation system, but no effect in paddy fields (Figure 6c).

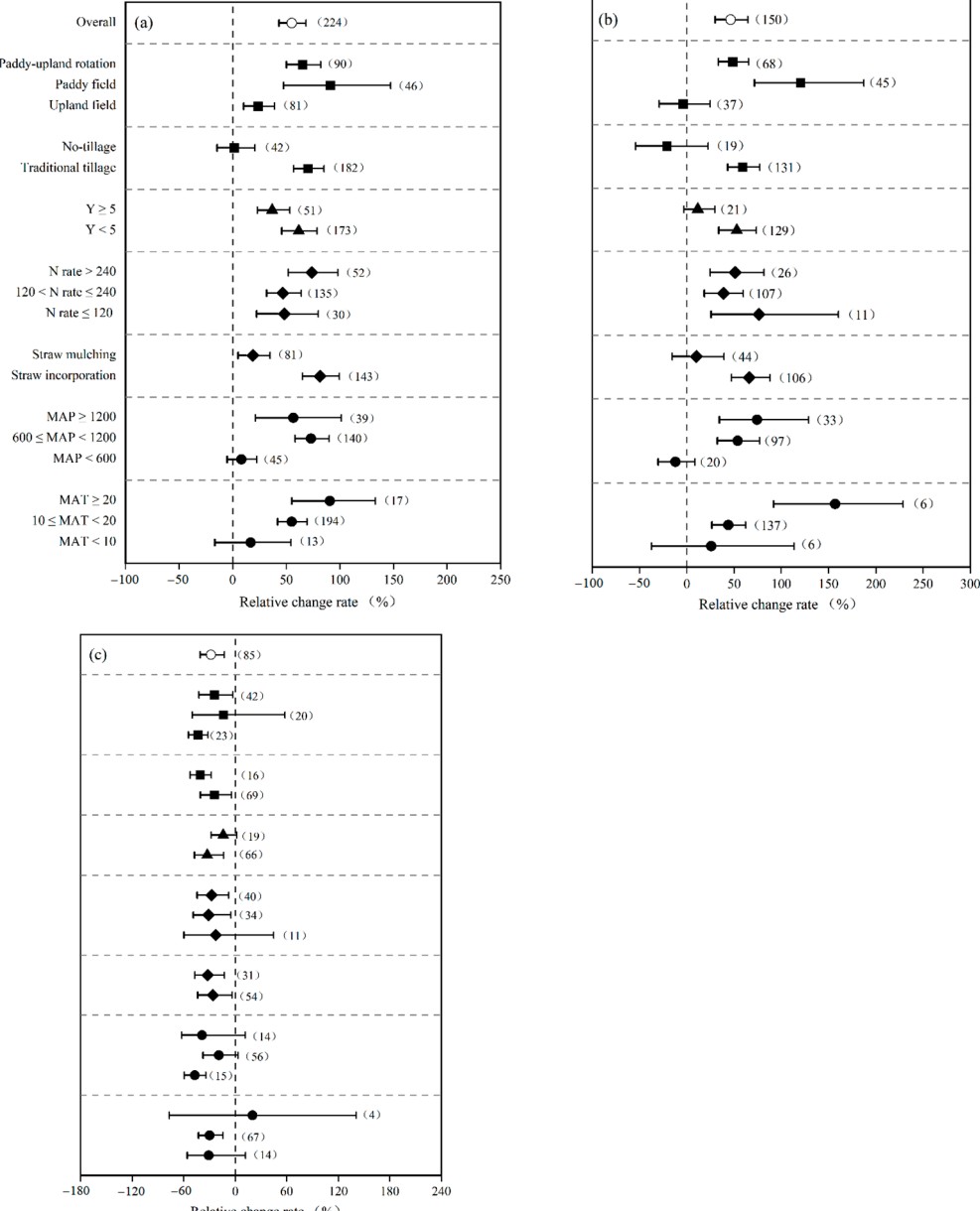

**Figure 6.** Changes in (**a**) global warming potential, (**b**) greenhouse emission intensity, and (**c**) net global warming potential under straw input to cropland.

## 4. Discussion

### 4.1. Impacts of Straw Input on Soil Carbon and Nitrogen Components

Our global meta-analysis of 1427 comparisons found that straw input increased nutrient supply. Indeed, the beneficial effects of straw amendment on soil quality received widespread attention [32,33]. Previous research found that straw input improves nutrient supply and enhanced SOC pool [34,35]. In our global meta-analysis, straw input increased SOC content by 13.25%, which was comparable to previous studies that increased SOC by 12.8% to 13.97% [17,36]. The positive effects on nutrient supply under straw input were attributed to supplemental nutrient inputs in the incorporated straw, in addition to the recommended mineral fertilizer rate [37], which decreased mineral nitrogen ($NH_4^+$-N, $NO_3^-$-N) loss by soil physical adsorption [38]. The increment of SOC content was increased as the amounts of nitrogen fertilizer application increased, N availability can also limit soil microbial activity, which could decrease the rate of organic C decomposition, leading to increased soil C content [39,40]. The increase in SOC content is greater with straw C/N ratio $\geq$ 60 (Figure 2a), straw with larger C/N ratio is rich phenolic/lignin compounds, and these substances act as binding agents for the formation of soil aggregates, which promote SOC accumulation [41]. High levels of MAT ($\geq$20 °C) resulted in relatively greater SOC content, which was potentially a consequence of high microbial activity, as well as the fast decomposition rate of straw carbon under a high level of MAT [42]. The duration of the experiment is also an important factor influencing the SOC content [2]. The SOC content increased with experiment duration, and soil C saturation was observed with straw input over 10 years [2,15]; however, the time of "C saturation" under straw input was not indicated in the dataset presented herein. Our findings show that no-tillage combined with straw input provides more obvious benefits in C sequestration, as demonstrated by previous studies [43,44], by providing a favorable environment for crop residue humification and increasing soil aggregate stability, which reduces the accessibility of SOC for oxygen, enzymes, and microorganisms, thereby reducing the mineralization of SOC but enhancing the stabilization rate of crop residues [45,46]. Our findings show that the increment of SOC content was lower in paddy fields than that in upland fields (Figure 2a), which is consistent with previous results [14]; it is possible that paddy soils have higher C levels initially, a negative relationship between SOC sequestration rates and initial C stocks, as well as a lower saturation deficit [47]. In addition, due to the influence of soil conditions, such as moisture, aeration, and temperature under periodic wet-dry cycling, the soil C decomposition rate was increased under paddy–upland rotation [48], and increasing soil heterotrophic respiration and decreasing crop stubble carbon input under paddy–upland rotation resulted in a significant C loss [49]. Thus, the effect of SOC increment under paddy–upland is lower than in continuous flooded paddy fields or upland fields (Figure 2a).

The active SOC fractions, including DOC and MBC content, were higher than the total SOC content response to straw input. The reason was possibly that straw input mainly provided readily decomposable substrates to the soil, and the active components of SOC responded more sensitively to straw input [14,50]. Furthermore, MBC and DOC content were found to be significantly related to SOC content (Figure 7). Previous reports also found that higher SOC leads to greater production of active SOC components [51,52]. Among the findings presented here, the response effect on the soil nitrogen component was altered by straw input rate and manner, climatic conditions, and farming regimes (e.g., nitrogen fertilizer rate, land use type and straw input duration; Table 1, Figure 2). Straw input can not only prevent surface runoff, but also slow down the infiltration of rainwater into the soil, limit the movement of nitrogen, and improve available nitrogen content [53]. Particularly, straw input increased AHN content at low and middle nitrogen fertilizer rates, but not at high nitrogen fertilizer rates (Figure 2e), which may be due to excessive nitrogen fertilizer causing imbalances in nutrient stoichiometry, and thus negatively affecting microbial biomass and diversity [54].

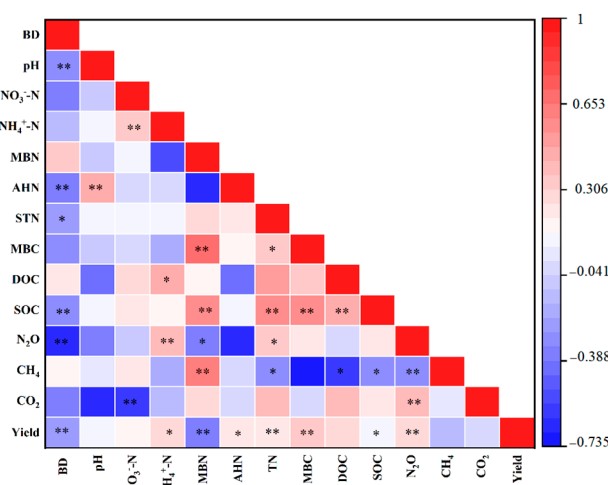

**Figure 7.** Heatmap of correlations of liner relationships between lnR of indicators responding to straw input. Abbreviations: $CO_2$, $CO_2$ cumulative emissions; $CH_4$, $CH_4$ cumulative emissions; $N_2O$, $N_2O$ cumulative emissions; and BD, soil bulk density. The symbols * and ** denote significant at the 0.05 and 0.01 probability level, respectively.

*4.2. Impacts of Straw Input on Crop Productivity and Nitrogen Fertilizer Use Efficiency*

Straw input improved soil structure, moisture retention and fertility, stimulated the metabolic activity of microorganisms, and reduced soil nitrogen loss, resulting in better crop growth conditions [55]. The benefits of straw input for increasing crop productivity are becoming more widely recognized [11,56]. Crop yield could increase by 8.86% after straw input, which is comparable to the 8.10% increase reported in China [11], but less than the 12.80% increase reported by others [14]. Crop yield is the result of multiple factors, the most important of which are straw input rate, nitrogen fertilizer rate, and MAT (Figure 3a). Crop yields were highest when straw mulching rate ranged from 4.5 to 9 t ha$^{-1}$, the dosage of straw mulching affected crop emergence rate and seedling growth, and although straw mulching was beneficial to crop growth in the early growth stage, it caused water stress in the late growth stage [57,58], which supported our hypothesis. Crop yield was significantly increased with the input duration (Figure 3), but the relatively highest increment of crop yield was observed under the duration of 5–10 years, which is consistent with previous research [15].

Furthermore, straw application is shown to significantly increase NUE, which is consistent with previous research [59]. That could be because straw input can replace a portion of chemical fertilizer application, improve soil nitrogen bioavailability, and reduce nitrogen loss. On the other hand, higher SOC content under straw input also facilitates a better synchronization between crop nutrient demand and soil nutrient supply, which can promote crop N uptake [17,41]. The increment effect of NUE influenced by straw incorporation rate (Figure 3b), which could be due to the fact that low straw rate returned to the field, is easier to decompose by soil organism, and mineral nitrogen released during decomposition can be directly used and absorbed by crops to improve net nitrogen mineralization, whereas high straw incorporation rate causes a conflict between straw decomposition and crop N utilization, resulting in soil available nitrogen deficiency [60,61]. Straw input significantly improved NUE at low and middle nitrogen application rates, with the greatest weighted response ratio found with a nitrogen application rate of <120 kg ha$^{-1}$ (Figure 3b), straw input reduced the amount of nitrogen fertilizer used and then improved NUE [62], and excessive nitrogen input exceeded plant growth demand, resulting in low nitrogen consumption efficiency; thus, high nitrogen application rate wastes soil resources and harms sustainable agricultural development in the long run [63].

### 4.3. Impacts of Straw Input on Greenhouse GASES emissions

Straw input significantly increased cumulative $CO_2$, $CH_4$, and $N_2O$ emissions (Figure 4), accelerating soil C and N turnover. The data presented herein show that $CO_2$, $CH_4$ and $N_2O$ significantly increased by 24.81%, 79.30%, and 28.31%, respectively, which was comparable to previous studies in China's croplands that increased $CO_2$, $CH_4$, and $N_2O$ by 31.7%, 130.9%, and 12.2%, respectively [15]. Previous studies well documented the priming effect mechanism under straw input, owing to increased soil microbial biomass and providing a good habitat for soil microbes that accelerate utilization of native soil organic matter and new added substrate [64]. On the other hand, root autotrophic respiration also increases with increased crop productivity [65].

Global subsoil organic carbon turnover times are dominantly controlled by climate. Subsoil SOC is very sensitive to warming and can significantly contribute to accelerated $CO_2$ emissions, which was well documented by Pries et al. [66]. Increasing the temperature and precipitation accelerated the $CO_2$ emissions under straw amendment (Figure 4a), which was also illustrated by previous studies [53]. The cumulative $CO_2$ emissions from upland soil were higher than those from paddy soil, which contradicted previous research [14]. This could be because more carbon matrix was contributing to $CH_4$ emissions from paddy soil under straw input [67]. Furthermore, our global meta-analysis found that no-tillage can reduce $CO_2$ and $CH_4$ emissions when compared to traditional tillage with straw input (Figure 4a,b), this occurs because no-tillage creates a favorable environment for crop residue humification [68], increasing the rate of plant biomass carbon stabilization [69]. Then, it increases soil aggregate stability and reduces SOC accessibility to oxygen, enzymes, and microorganisms, reducing SOC mineralization and $CO_2$ emissions [46,70]. Moreover, partial straw is aerobically degraded on the soil surface, and its degradation products are less likely to be converted to $CH_4$ in the soil oxide layer [71]. The effect of long-term straw input on $CO_2$ and $CH_4$ emissions was typically lower than that of initial straw input, which was consistent with previous studies [15]. Long-term straw input alters soil structure and quality, particularly soil aggregate, which may protect SOC from decomposition, resulting in reduced $CO_2$ and $CH_4$ emissions [15]. Increased soil fertility from long-term straw incorporation can also promote algal growth, which increases dissolved $O_2$ concentrations [72]. Moreover, larger plants promote $O_2$ transport into the rhizosphere, which stimulates methanotrophic growth [73].

Our global meta-analysis showed that straw input significantly influenced $CH_4$ emissions (Figure 4b), which was comparable to the previous study [11,15]. The incremental effect of $CH_4$ emission decreased with the straw input rate, despite the fact that a high rate of straw incorporation provides a rich substrate for methanogens, aerobic degradation of straw, and substrate for C and N reactions for nitrification and denitrification results in increased $CO_2$ emissions but no $CH_4$ emissions [74]. Straw input increased $CH_4$ emissions by 211.90% in the paddy fields, but not in the upland fields (Figure 4b). Flooding or drainage in paddy fields might result in greater MBC content, a faster shift in soil physical and chemical properties, as well as microbial community abundance and structure, and thus more greatly accelerate C turnover and stimulate $CH_4$ release under straw input, as previous research showed [75]. For the paddy–upland rotation, $CH_4$ emissions increased by 102.89%, 98.99%, and 19.93% under annual straw input, straw input in rice season and straw input in dry season, respectively (Table 2). After aerobic decomposition in dry farming season, the organic carbon in organic materials mostly exists in the form of macromolecular compounds, which has little effect on $CH_4$ emissions [76]. As a result, shifting straw application from the rice season to the non-rice season can effectively avoid high $CH_4$ emissions [74]. There is a trade-off between $CH_4$ and $N_2O$ emissions caused by agricultural practices [77,78]. $N_2O$ emissions increased by 22.59%, 1.01%, and 13.73% under annual straw input, rice season straw input, and dry season straw input, respectively (Table 2). Straw input improves soil water and temperature, as well as creating a more anaerobic environment through straw decomposition, and consumes oxygen, which promotes nitrification and denitrification and stimulates $N_2O$ production from upland crops [10,19]. On the other hand, increased

carbon to nitrogen ratio leads to full utilization of nitrogen sources by microorganisms and reduces $N_2O$ emissions from paddy fields [79], therefore, optimizing straw application time and straw input manner is critical for balancing $CH_4$ and $N_2O$ emissions.

**Table 2.** Effect of straw input in different seasons on $CH_4$, $N_2O$ cumulative emissions and GWP for paddy–upland rotation field.

| | Mean ($CH_4$ Emissions) | CIs | Mean ($N_2O$ Emissions) | CIs | Mean (GWP) | CIs |
|---|---|---|---|---|---|---|
| Straw input in dry season | 17.93 | −27.83 to 91.02 | 13.73 | −5.46 to 34.54 | 16.88 | −26.25 to 83.20 |
| Straw input in rice season | 98.99 | 57.98 to 148.88 | 1.01 | −9.64 to 14.01 | 53.82 | 28.56 to 83.31 |
| Annual straw input | 102.89 | 72.82 to 140.54 | 22.59 | 8.78 to 39.43 | 62.04 | 45.34 to 83.14 |

Note: The number of experimental observations under straw input in different seasons for upland–paddy rotation are 11, 53, and 39 for $CH_4$; 24, 50, and 45 for $N_2O$; 10, 50, and 39 for GWP.

Overall, straw input increased soil $N_2O$ emissions (Figure 4c), which is consistent with previous research [80]. Soil $N_2O$ emissions were significantly affected by straw input rate, which is also reported by Xia et al. [17]. Straw input regulates soil temperature, water-filled pore space, and consumption of soil $O_2$ from enhanced organic matter decomposition. High straw input rate may contribute to a greater prevalence of anaerobic conditions that favor denitrification [10]. The cumulative $N_2O$ emissions showed a positive regression relationship with Triticeae and Fabaceae straw input rate, and Triticeae and Fabaceae straw with a lower C/N ratio can be decomposed quickly, leading to higher nitrogen availability for nitrification and denitrification [81]. Additionally, there is a significantly positive regression relationship between nitrogen fertilizer rate and cumulative $N_2O$ emissions under maize and rice straw input. Straw with high C/N ratio increased nitrogen consumption and caused more nitrogen immobilization during straw degradation. Fertilization to remedy for the lack of nitrogen, reduces nitrogen fixation would accelerate $N_2O$ emissions [82]. Studies showed that crop residues with a C/N ratio lower than 20–30 are expected to cause net nitrogen mineralization due to their high nitrogen concentration, while those with C/N ratios higher than 30 were found to result in net nitrogen immobilization [83].

*4.4. Impacts of Straw Input on Global Warming Potential*

Soil carbon sequestration is more effective as a climate change mitigation strategy [4,84], so the '4 per mille Soils for Food Security and Climate' initiative was launched at the COP21. The significant C sequestration potential of straw input combined with appropriate farming practices is critical for improving or maintaining current SOC stocks across all agroecosystems [34].

The GWP is commonly used to assess the potential effects of $CH_4$ and $N_2O$ emissions on the climate system in a comprehensive manner [82]. Our global meta-analysis showed that straw input significantly increased the overall GWP of $CH_4$ and $N_2O$ (Figure 6a). Furthermore, $CH_4$ emissions contribute more to GWP, GHGI and NGWP than $N_2O$ emissions (Figure 8a–f), which is consistent with previous research [2]. Therefore, the increase in GWP was more than twice as great in paddy fields as it was in upland fields (Figure 6a). The GWP can be reduced by integrating farming practices, particularly those aimed at reducing $CH_4$ emissions, such as straw mulching, no-tillage, straw input in upland fields, and straw input in the dry season for paddy–upland rotation (Figure 4b; Table 2). Furthermore, GHGI is an important parameter with agronomic and environmental implications. Our meta-analysis presented herein suggests the possibility of improving field management practices to reduce GHG emissions while increasing crop yield by balancing the trade-offs between climate change mitigation and global food security. Due to the low GWP under these conditions, straw mulching, straw input in upland field, no-tillage with straw input, and straw input in dry season all resulted in lower annual GHGI. This suggests that by optimizing straw management and farmland management practices, relatively high yields

and lower GWP can be achieved at the same time. Climatic regions greatly influenced the differences in GWP and GHGI, and higher temperature changed in soil physicochemical and biological properties can promote $CH_4$ emissions by increasing C or N availability and microbial activity [85], higher precipitation frequency, combined with greater soil moisture promote denitrification-driven $N_2O$ emissions and conducive to $CH_4$ generation [21]. The balance between SOC storage and $N_2O$ and $CH_4$ emissions is used to calculate NGWP, which reflects a complete understanding of agriculture's impact on radiative forcing [86]. With effective management practices, soil carbon storage can offset GHG emissions from straw input [87]. Under straw input, the NGWP was significantly reduced depending on appropriate nitrogen fertilizer rate and land use type (Figure 6c).

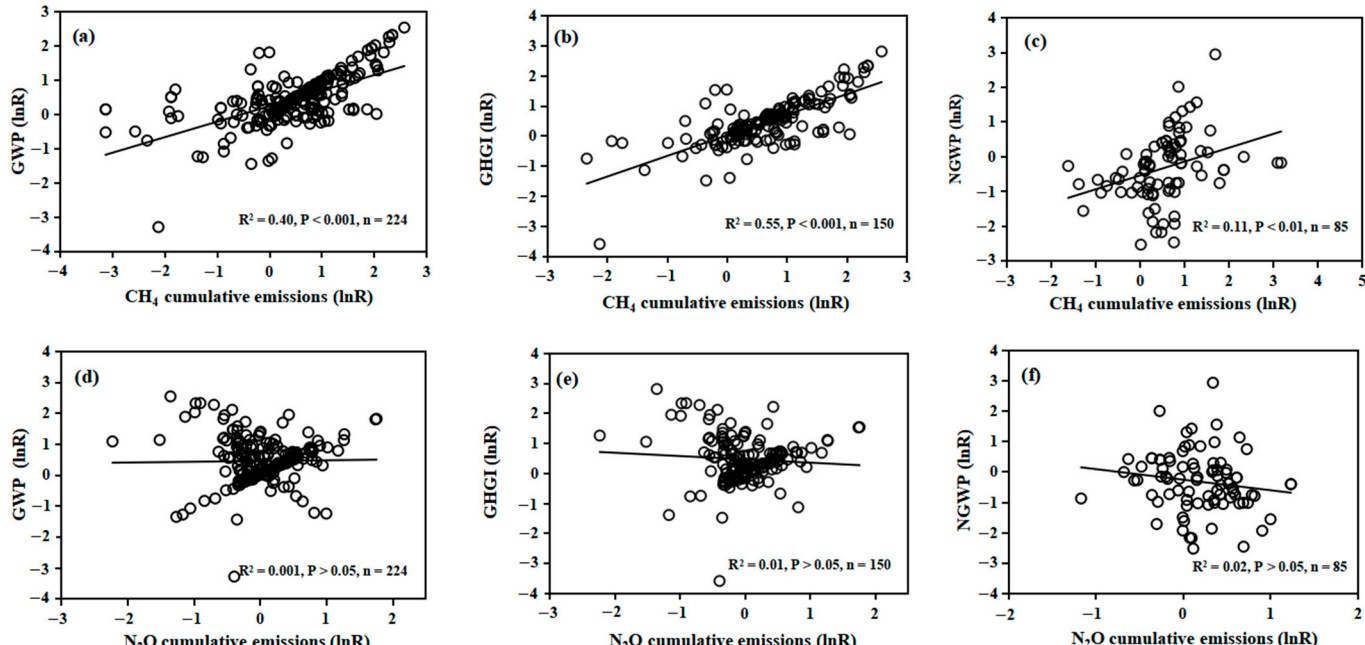

**Figure 8.** Relationship between the changes in $CH_4$ (**a**–**c**), $N_2O$ (**e**–**f**) cumulative emissions (lnR) and global warming potential (GWP, lnR), greenhouse emission intensity (GHGI, lnR), net global warming potential (NGWP, lnR) under straw input to global croplands. Negative values of lnR denote a reductive effect of straw input on GWP, GHGI, and NGWP, and n denotes number of comparisons.

## 5. Conclusions

Straw input is an environmentally friendly method of managing crop residues efficiently. Through studying the effects of different management practices under straw input on soil fertility, crop production, and GHG emissions, optimizing management practices under straw regimes for sustainable agricultural production is possible. This study showed straw input increased soil carbon and nitrogen components, crop yield, NUE, and GHG emissions in global agroecosystems. We identified promising measures to reduce GHG emissions, promotes crop growth and soil sustainability, which low straw input rate, straw mulching, long-term straw input, input of straw with C/N ratio > 30, and no-tillage are effective enhanced crop production and climate change mitigation strategies at a general level. Straw input benefits the most in regions with MAT < 10 °C and MAP < 600 mm. Moreover, straw should be returned preferentially during the dryland season in the paddy–upland rotation field to reduce the promoting effect on $CH_4$ emissions. The results of this meta-analysis provide both support and caution to straw input. Agricultural management practices, such as climatic conditions, straw input, and quantity and quality, nitrogen fertilizer rate, tillage method, and land use types should be adequately planned to help maintain soil quality, mitigate climate change, and ensure food security.

Overall, straw input improved agroecosystem function, production, and environmental benefits. Despite the fact that our findings provide clear information about the impact of straw application on soil properties, crop yield, NUE, and GHG emissions, our meta-analysis still has some flaws. Firstly, the limited dataset resulted in an uneven distribution of compiled observations across different regions in our meta-analysis, potentially increasing data uncertainty. Second, while it is acknowledged that soil type and soil basic condition have significant effects on GHG emissions and crop yield, our meta-analysis did not examine the magnitude of these effects due to variability and a lack of detailed information. Future research involving many factors in controlled experiments is required to fully understand the benefits of straw input to agricultural fields. These factors may include soil types and soil basic condition and crop species, etc., which would help resolve uncertainties observed in meta-analyses, and there may be some interactions between these factors that require in-depth research and exploration. It is also necessary to adjust field management practices and site-specific conditions in order to maximize the benefits of straw input.

**Supplementary Materials:** The following supporting information can be downloaded at: https://www.mdpi.com/article/10.3390/agronomy13030710/s1.

**Author Contributions:** P.L.: data curation, writing—original draft preparation, formal analysis. A.Z.: funding acquisition, conceptualization, reviewing and editing, project administration. S.H.: collected the data. J.H.: data curation, writing—review and editing. X.J.: collected the data. X.S.: collected the data. Q.H.: writing—review and editing. X.W.: writing—review and editing. J.Z.: writing—review and editing. Z.C.: writing—review and editing. All authors have read and agreed to the published version of the manuscript.

**Funding:** This work was supported by the National Key Research and Development Program of China (2021YFD1900700; 2017YFD0200106); And the Natural Science Foundation of Shaanxi Province, China (2020JQ-274).

**Institutional Review Board Statement:** Not applicable.

**Informed Consent Statement:** Not applicable.

**Data Availability Statement:** The data that support the findings of this study are openly available in the Science Data Bank archive at https://doi.org/10.57760/sciencedb.07323, accessed on 1 January 2023.

**Conflicts of Interest:** The authors declare that they have no known competing financial interests or personal relationships that could have appeared to influence the work reported in this paper.

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
