# Peer review of "Optimizing Management Practices under Straw Regimes for Global Sustainable Agricultural Production"

_agronomy, doi:10.3390/agronomy13030710_

Round 1

Reviewer 1 Report

Please read the attachment. thank you.

Reviewer 2 Report

This study has merit, and the overall results are interesting. Figures and writing should be improved.

L20: Please give the full name of these GHGs. Also, correct this in the introduction section. Define the abbreviations when it appears in the first place, and correct this throughout the manuscript.

L45: Straw retention could also improve soil's physical properties. Please cite this one. Residue retention and minimum tillage improve the physical environment of the soil in croplands: A global meta-analysis.

L57: Clarify the scale of this mate analysis.

L59: Nitrogen has been defined as N, then please consistently use N.

L63-64: How come mention the no-tillage here, this study focus on straw management.

L66-67: Consider reorganizing these arguments because C or N addition could induce SOM decomposition, and thus CO2 and N2O emissions as well. Organic matter contributions to nitrous oxide emissions following nitrate addition are not proportional to substrate-induced soil carbon priming. Please cite this one.

L78: were to (a) assess the over-----.

Clarify how the subgroups were made in the M&M.

The results section can be simplified, only need to highlight the important results instead of pointing out everything in the tables or figures.

Fig. 3-4. Better to add crop yield and NUE on the top of each subplot.

Figure 5. Better to use shapes to indicate GHG type as well. Shape pulse colour.

Figure 8. Keep two decimals for R2.

L410: Compare to which system in where? Give more details to comparisons.

L526: Better to mention the limitation of this study.

In conclusion, please give suggestions for future research.

Round 2

Reviewer 1 Report

Please read the attachment. Thank you.

Reviewer 2 Report

I appreciate the authors addressing my comments.

Author Response

We appreciate for Reviewer’ warm work earnestly.